# In Silico Prediction of Human Leukocytes Antigen (HLA) Class II Binding Hepatitis B Virus (HBV) Peptides in Botswana

**DOI:** 10.3390/v12070731

**Published:** 2020-07-06

**Authors:** Wonderful Tatenda Choga, Motswedi Anderson, Edward Zumbika, Bonolo B. Phinius, Tshepiso Mbangiwa, Lynnette N. Bhebhe, Kabo Baruti, Peter Opiyo Kimathi, Kaelo K. Seatla, Rosemary M. Musonda, Trevor Graham Bell, Sikhulile Moyo, Jason T. Blackard, Simani Gaseitsiwe

**Affiliations:** 1Research Laboratory, Botswana Harvard AIDS Institute Partnership, Gaborone 0000, Botswana; wchoga@bhp.org.bw (W.T.C.); manderson@bhp.org.bw (M.A.); bphinius@bhp.org.bw (B.B.P.); tmbangiwa@bhp.org.bw (T.M.); lbhebhe@bhp.org.bw (L.N.B.); kabobaruti@gmail.com (K.B.); kseatla@bhp.org.bw (K.K.S.); rmusonda@bhp.org.bw (R.M.M.); smoyo@bhp.org.bw (S.M.); 2Division of Human Genetics, Department of Pathology, Faculty of Health Sciences, University of Cape Town, Cape Town 7925, South Africa; 3Department of Applied Biology and Biochemistry, Faculty of Applied Sciences, National University of Science and Technology, Bulawayo 0000, Zimbabwe; edward.zumbika@nust.ac.zw; 4Department of Biological Sciences, Faculty of Science, University of Botswana, Gaborone 0000, Botswana; 5Centre for Proteomic and Genomic Research, Cape Town 7925, South Africa; popiyo@aims.ac.tz; 6Department of Medical Laboratory Sciences, Faculty of Health Sciences, University of Botswana, Gaborone 0000, Botswana; 7Department of Immunology and Infectious Diseases, Harvard T.H. Chan School of Public Health, Boston, MA 02115, USA; 8Independent Researcher, P.O. Box 497, Wits, Johannesburg 2050, South Africa; trevorgrahambell@gmail.com; 9Division of Digestive Diseases, University of Cincinnati College of Medicine, Cincinnati, OH 45267, USA; jason.blackard@uc.edu

**Keywords:** hepatitis B virus (HBV), HLA class II alleles, T-cell epitopes, in silico, immunoinformatics, candidate multi-epitope vaccines (MEV), escape mutation, Botswana, Africa

## Abstract

Hepatitis B virus (HBV) is the primary cause of liver-related malignancies worldwide, and there is no effective cure for chronic HBV infection (CHB) currently. Strong immunological responses induced by T cells are associated with HBV clearance during acute infection; however, the repertoire of epitopes (*epi*) presented by major histocompatibility complexes (MHCs) to elicit these responses in various African populations is not well understood. In silico approaches were used to map and investigate 15-mers HBV peptides restricted to 9 HLA class II alleles with high population coverage in Botswana. Sequences from 44 HBV genotype A and 48 genotype D surface genes (*PreS/S*) from Botswana were used. Of the 1819 *epi* bindings predicted, 20.2% were strong binders (SB), and none of the putative *epi* bind to all the 9 alleles suggesting that multi-epitope, genotype-based, population-based vaccines will be more effective against HBV infections as opposed to previously proposed broad potency epitope-vaccines which were assumed to work for all alleles. In total, there were 297 unique *epi* predicted from the 3 proteins and amongst, S regions had the highest number of *epi* (n = 186). Epitope-densities (D_epi_) between genotypes A and D were similar. A number of mutations that hindered HLA-peptide binding were observed. We also identified antigenic and genotype-specific peptides with characteristics that are well suited for the development of sensitive diagnostic kits. This study identified candidate peptides that can be used for developing multi-epitope vaccines and highly sensitive diagnostic kits against HBV infection in an African population. Our results suggest that viral variability may hinder HBV peptide-MHC binding, required to initiate a cascade of immunological responses against infection.

## 1. Background

Hepatitis B virus (HBV), a member of the *Hepadnaviradae* family, is the major etiology of end stage liver diseases (ESLD), liver cirrhosis (LC) and hepatocellular carcinoma (HCC), and causes up to 887,000 deaths per year [1]. Although more than 90% of healthy adults resolve acute HBV infection within 6 months, there remain over 287 million people who test seropositive for hepatitis B surface antigen (HBsAg) [2] and have chronic HBV infection (CHB). Viral clearance is mediated by cytokines, lymphocytes, and the ability to mount a multi-specific polyclonal and vigorous T cell-mediated response against HBV antigens for a protective immunity [3,4,5]. The quality of these responses is influenced by host genetics, as well as the ability of certain viral variants to escape immune recognition [6,7,8].

The major histocompatibility complexes (MHCs)—known as human leukocytes antigens (HLAs) in humans—are integral components of host genes located at chromosome *6p21.* These highly polymorphic proteins serve as mediators of adaptive immune responses by presenting processed antigenic peptides to T cells. The two compatible types of MHCs—class I and class II—present exogenous and endogenous epitopes to CD8^+^ cytolytic T cells and CD4^+^ T helper (T_h_) cells, respectively [9]. The MHC class II alleles (HLA-DR, -DQ and -DP) present epitopes to CD4^+^ T cells [*epi*-HLA class II → CD4^+^ T-cells] that in turn elicit adaptive immune responses against viral infections by facilitating the induction of CD8^+^ cytotoxic T-lymphocytes (CTLs), production of cytokines crucial for survival, and maturation of B cells [10,11,12,13].

The link between HBV pathogenesis and host immunological profiles is still poorly understood [2]. HBV exhibits a high mutation rate, although only a small number of amino acid substitutions have been characterized functionally due to the costly and time-consuming nature of in vitro assays. Recent approaches have utilized in silico approaches such as machine learning techniques to prioritize candidate peptides for in vitro assays [14,15]. Thus far, the HBV mutations characterized have been associated with sensitivity of immunologic and molecular-based assays and viral escape leading to poor prognosis [16,17,18,19]. However, amino acid variations that influence HBV-MHC binding are poorly understood. In silico mapping of HLA class II binding peptides can be used to identify candidate peptides for in vitro assays to confirm CD4^+^ T cell epitopes which are crucial for the design of epitope-based vaccines and highly sensitive diagnostic tools that can detect low HBV DNA levels which are frequently missed by diagnostic kits [20,21,22,23].

Sub-Sahara Africa (SSA) and the Western Pacific regions are highly endemic for CHB [24], where the circulating genotypes include A, D, and E for SSA, and B and C for the Western Pacific. HBV genotypes A (subgenotype A1) and D (subgenotype D3) have been reported in Botswana, with genotype E occurring rarely [25,26,27,28,29]. Not only do genotypes show unique geographic distribution, they also differ in treatment response, pathogenesis potential, and prognosis [30,31]. Studies conducted in China have mapped different T cell epitopes that may be eligible for epitope-based vaccines and some were evaluated in vitro [20,32]. However, these findings may be less applicable to African populations whose host genetic pool, circulating genotypes, and immune profiles for HBV (e.g., hepatitis B e antigen [HBeAg] and HBsAg positivity) differ considerably with those of the Chinese population. A prerequisite to determine the epitope(s) for inclusion in epitope-based vaccines include (1) identification of conserved regions of the genome and (2) characterization of those regions that elicit protective immune responses [33]. Although there are hepatitis B vaccines in use currently, vaccine escape does occur; thus, more optimized vaccine candidates may be needed to avoid vaccine failure. In this study, we utilized HLA class II alleles that occur at the highest frequency in Botswana and locally derived HBV strains to identify HLA class II binding peptides which are good candidates for confirmatory in vitro tests of immunogenicity. The present study had three major aims: (1) to determine the repertoires of HLA class II epitopes within HBV envelope sequences of genotypes A and D isolated from different risk groups in Botswana (described in our earlier papers); (2) to compare if the predicted epitopes in genotypes A and D may vary across other HBV genotypes, suggesting that genotype-based multi-epitope vaccines would be more successful than the broad potency vaccines currently in use; (3) to investigate amino acid variations within these epitopes to determine if they may lead to immune escape (i.e., candidate escape mutations).

## 2. Materials and Methods

### 2.1. Mapping Peptides from HBV Surface Gene Restricted to HLA-class II Alleles

Three sequence datasets were included in the current analysis. The first database (N_1_) was used to map epitopes (*epi*) that bind predominantly to HLA class II alleles in Botswana and consisted of 92 non-recombinant full-length S gene (*PreS/S*) sequences [25,26,28] retrieved from GenBank (accession numbers MF979142—MF979176, KR139743—KR139748, and MH464807—MH464854). The aligned sequences were sorted by genotype and included A (n = 44) and D (n = 48). The three domains of HBV surface proteins—PreS1, PreS2, and S—were manually extracted from an overlapping *Pol/S* fragment and were divided into genotypes whose amino acid (aa) sequence alignments were sorted according to column similarities. Nucleotide alignments and sorting were performed using AliView 1.21 software [34]. Each region was then used to create a consensus sequence with the threshold set at 90% for all positions. Variants that did not meet this threshold were investigated independently in post-analyses. To assess if the aa composition of consensus sequences was representative of existing HBV strains, BLAST searches were conducted using the NCBI database, and strains exhibiting 100% similarity and coverage were evaluated further (Appendix A).

The 15-mer HBV peptides overlapping by 14 aa were tested for binding to 9 HLA class II alleles—HLA-DRB1*0101, DRB1*0301, DRB1*0401, DRB1*0701, DRB1*0802, DRB1*1101, DRB1*1302, DRB1*1501, and DRB5*0101—that have high population coverage in Botswana [35]. The NetMHCIIPan version 3.2 online server (http://www.cbs.dtu.dk/services/NetMHCIIpan/) [36] was used to predict binding peptides, and their binding-affinity scores were categorized based on the Log-transformed binding affinity [1-Log50k (aff)]. The settings were adjusted starting with default [1-Log50k (aff)] of 0.426; 500nM and logAff = 0.638; 50nM for weak and strong binding respectively. Ten previously characterized HBV envelope proteins (PreS1, PreS2, S) epitopes—S: 18–37 epitope ID 51310; S: 70–84 (3966); S: 83–98 (46959); S: 200–211 (66307); S: 201–215 (59353); S: 211–244 (6574); S: 230–247 (47877); S: 363–378 (76458); S: 376–389 (17331); S: 378–386 (37664)—available at (www.iedb.org) [37] were included to calibrate the tool’s settings, and the thresholds showing highest specificity were utilized in the present study. Using percentile rank of eluted ligand prediction score (%Rank_EL), the strong binders (SB) were determined between 2–10% Rank_EL and 10–50% Rank_EL for weak binders (WB). Peptides with binding affinity less than that of WB were deemed pseudo binders (NB). Figure 1 outlines the various analytical steps included in this study, and the NetMHCIIPan results are provided in the Appendix A.

The relationship between the length of the protein and the frequency of binding-peptides were compared using epitope density score (D_epi_) for all protein domains of S gene (PreS1, PreS2, S) and genotypes (A versus D). D_epi_ score was defined as the proportion of binding peptides $\sum_{i=1}^{n}I$_epi_ (where; ***I***_epi_ = WB + SB) to total predicted epitopes (T_epi_) relative to protein size. T_epi_ represent the total count of all predicted proteins.

### 2.2. Determining Prevalence of Putative Epitopes in HBV Genotypes (A–I) Except A and D

Putative or promiscuous *epi* were defined as peptides that exhibit similar binding affinity to 2 or more alleles. The prevalence of predicted putative *epi* were determined in a second dataset (N_2_) that included 10,308 PreS/S sequences (genotype B = 2905; C = 5575; E = 1118; F = 477; G = 86; H = 69; I = 78) retrieved from HBV database available at http://hvdr.bioinf.wits.ac.za/alignments/index.html [38]. N_2_ was also used to determine the overall prevalence of the predicted escape mutations. The sequences used in this analysis are included in the Appendix A provided; Appendix A.

### 2.3. Variations Causing Escape to HLA Class II Binding

Dataset N_3_ consisted of 7743 HBV sequences (genotype A = 3115; genotype D = 4628) used to determine the frequency of aa variations which were termed *HLA escape mutations* in other HBV sequences. Sequences used were curated from http://hvdr.bioinf.wits.ac.za/alignments/index.html, partitioned by proteins (PreS1, PreS, S) [38]. Escape mutations were defined as those aa variations within the 15-mer core aa sequence that cause the binding affinity to change from either strong to pseudo binding (SB → NB) or from weak to pseudo binding (WB → NB). Several in-house customized Python pipelines were used to accurately investigate the frequency of escape mutations. The sequences used in this analysis are in the Appendix A provided; Appendix A.

### 2.4. Screening Putative Epi and Reconstruction of Tertiary Structure of the Modelled Vaccine

Since the predicted putative *epi* can be also homologous to human peptides that may (1) cause either autoimmune responses when used as a vaccine or (2) give false results when used as a diagnostic marker, a BLAST search was conducted with the NCBI protein database for all immunogenic *epi*. Afterwards, the predicted putative *epi* were catenated using a previously described method [39], and different combinations were used to construct candidate multi-epitope vaccines (MEVs). Physiological and biochemistry proteins such as thermal stability, desirable shelf-life, and pH among other properties are prerequisites during development of an ideal vaccine. The biochemical properties of generated candidate proteins were evaluated using online ProtoParam tool [40]. The proteins exhibiting properties similarly to those of vaccines currently in use were deemed the best candidate. The properties of 3 current HepB vaccines—including VO_0011094, VO_0011095, and VO_0011093—are curated in the DNA vaccine database [41]. Properties predicted include; immunogenicity, antigenicity, instability index, estimated half-life in humans’ molecular weight (mw), aliphatic index (AI), grand average of hydrophobicity (GRAVY), and theoretical (pH).

### 2.5. Determining the HLA-HBV Association Using T_epi_

To test the hypothesis that T_epi_ could serve as a useful predictor of HLA-HBV associations, T_epi_ of S genes (A and D) were used to rank the 9 HLA class II alleles in the post *epi* prediction analyses. The available literature was used to corroborate the analyses.

## 3. Results

### 3.1. Predicting T-cell Epitopes Using Consensus from N_1_ Dataset 

We first generated 6 different consensus sequences from *PreS/S* sequences of genotypes A and D. Consensus sequences were validated by comparison to the NCBI database and identified 16 pre-existing sequences which exhibit 100% identity to: *PreS1* genotype A (PreS1_A_) consensus sequence; 39 to PreS1_D_, 6 to PreS2_A_, 12 to PreS2_D_, 19 to S_A_, and 20 to S_D_, respectively. The consensus sequences, identical sequences, and their country of origin are provided in Appendix A; Appendix A. *Epitope* is defined as a peptide that binds either weakly or strongly bind to HLA-DR alleles used, while *HLA class II escape mutations* were defined as aa variations within the 9-mer core aa sequence that changed the epitope-HLA-DR binding from SB/WB to pseudo binding. In total, there were 1819 total binding predicted (T_epi_) from 297 unique epitopes restricted to 9 HLA class II including 20.2% SB. The number of signature aa differentiating HBV genotypes A and D were 31, 13, and 9 for the S, PreS1, and PreS1 regions, respectively (Table 1). S protein had the highest binding peptides constituting 79.9% of the sum of all T_epi_ (∑T_epi_), PreS1 constituted 11%, while PreS2 had the least (9.1%) ∑T_epi_. The D_epi_ of the SB and WB among genotypes (A and D) and proteins (*PreS1*, *PreS2*, *S*) are summarized in Figure 2.

### 3.2. Prevalence of Putative Epitopes Across all HBV Genotypes 

To assess if the predicted *epi* may be suitable for vaccine inclusion, the most common *epi* were compared to sequences other than genotypes A and D. Table 2 shows the number of aa sequences containing the indicted *epi*. Eight out of 53 *epi* were found in semi-conserved regions ranked as “++++” and had prevalence > 85% among sequences in N_2_ dataset. These results suggest that multi-epitope genotype-based vaccines may be better to avoid vaccine escape.

### 3.3. Profiles of Strong Binding Epitopes

SB *epi* of the 3 proteins were sorted by the core aa sequence and analyzed based on genotype. Those found in both genotypes were considered putative when binding to alleles. There were no SB epitopes that were common between sequences of genotypes A and D for PreS1 and PreS2 regions. S protein had 89 out of 230 *epi* that satisfied the above criteria and were promiscuous for at least 5 alleles. There were 5 unique *epi* whose core aa were at S protein residues 41–49 (FLGGSPVCL), 14 *epi* with S protein residues 20–28 (FLLTRILTI), 10 *epi* with S residues 183–191 (FVGLSPTVW), 9 *epi* with S residues 22–30 (LTRILTIPQ), 6 *epi* with S residues 162–170 (LWEWASARF), 6 *epi* with S residues 184–192 (VGLSPTVWL), 12 *epi* with S residues 96–104 (VLLDYQGML), 2 *epi* with S residues 180–188 (VQWFVGLSP), 9 *epi* with S residues 163–171 (WEWASARFS), 4 *epi* with S residues 182–190 (WFVGLSPTV), and 12 *epi* with S residues 72–80 (YRWMCLRRF). Table 3 shows the full profiles of SB *epi* mapped for the S proteins for genotypes A and D.

### 3.4. Profiles of Most Promiscuous Epitopes

Since the majority of predicted *epi* were WB, a strict threshold was applied to select the most the promiscuous *epi*. Thus, *epi* were selected if they bind to least 6 alleles or more as shown in Table 4. 

### 3.5. Designing and Predicting Structure of Candidate Multi-Epitope Vaccine

Table 4 shows 27 putative *epi* binding to at least 6 alleles and were selected to model the tertiary structure of candidate vaccines. The overlapping S region *epi* (S_A_ = 11, S_D_ = 10) were catenated to form proteins which were used to model a tertiary structure as shown in Figure 3. The overlapping S_A_
*epi* at S protein residues 6–20 with aa sequence SGFLGPLLVLQAGFF, aa sequence CIPIPSSWAFAKYLWEWASVRFSWLSLLVPFVQWF at S protein residues 155–183, and aa sequence WYWGPSLYNILSPFIPLLPIFFCLW at S protein residues 199–223 yields a 75 amino acid protein of mw = 8878.62. Of the 6 proteins predicted, the most stable S_A_ protein was vacci-S_A_ with amino acid sequence: 5′-SGFLGPLLVLQAGFFWYWGPSLYNILSPFIPLLPIFFCLWCIPIPSSWAFAKYLWEWASVRFSWLSLLVPFVQWF-3′. The overlapping *epi* in S_D_ occupied residues S: 6–20, S: 68–82, S: 197–223 and S: 155–183. The resulting protein was 85 aa long and had mw = 10,749.99. Of the 24-proteins predicted, the one selected for constructing tertiary structure was vacci-S_D_ with aa sequence: 5′-MMWYWGPSLYSILSPFLPLLPIFFCLWSGFLGPLLVLQAGFFSWAFGKFLWEWASARFSWLSLLVPFVQWFTCPGYRWMCLRRFIIFLF-3′. When a BLAST search was conducted with the NCBI protein database for the 2 protein sequences, results show that both sequences were similar to 2 domains of major surface antigen (vMSA) from hepadnavirus superfamily; accession number pfam00695. A similar approach was used to select proteins from the *epi* in PreS2 region to determine proteins that can be used to model the 3D tertiary structures of *epi* in the PreS2 region. In total, there were 6 *epi* selected 3 for each genotype. The overlapping *epi* were 20 aa long with mw of 2006.20 and occupying residues PreS2: 34–53 in genotype A sequences. vacci-PreS_D_ was made from overlapping *epi* occupying residues PreS2: 37–53 and had a mw of 1719.91. All proteins generated using the different epi ordering (permutations and combinations) have been provided under the Appendix A.

### 3.6. Mutations Associated with Escape from Class II HLA Binding

For amino acid variants present in more than 5% of sequences of N_1_ database, we evaluated the in silico impact on immune recognition. Mutations were labelled relative to the HBV surface gene proteins—PreS1_1–115_, PreS2_1–42_, and S_1–226_. Mutations that hindered HBV peptide-HLA binding were 86T, 90T, and 94P in PreS1_A_; 54P, 79E, 84S, and 85Q in PreS1_D_; 12I, 31I, and 54P in PreS2_A_; 5F, 22H, 22L, 22P, 32H, 36L, and 42S in PreS2_D_. The list of escape mutations and their prevalence in 7743 HBV genotype A and D sequences are shown in Table 5 and supplementary file Appendix A. There were coordinated variations among positions in aa sequence alignments that had an impact on HBV-HLA binding but showed no impact when analyzed individually. These were termed covariance mutations. The combinations of the covariance mutations and their impact on binding potential are shown in Table 5, Table 6 and Table 7 below.

### 3.7. Assessing the Distribution of HBV-HLA (epi) as a Predictor of HLA Protective Effective

We used existing information on immunological studies for meta-analysis to estimate T_epi_ as a predictor of the protective effect of MHC class II alleles. Figure 4 shows a Pareto analysis and the 20% threshold indicates the 3 HLA class II alleles—DRB1*0301, 1302, *1101—that are highly likely to be less protective against HBV. We therefore speculate, with caution, that there is a relationship between T_epi_ and protectiveness and this should be further investigated to establish correlations (*p*-value < 0.05) with high statistical confidence.

## 4. Discussion

This detailed HBV immunoinformatics approach outlines the candidate peptides that can be used to develop biologicals against CHB. However, mutations within the T cell epitopes may impair HBV-HLA complex formation, a crucial component responsible for initiating cascade of responses for viral clearance [67]. In this study, we observed that there were no *epi* that bind to all alleles, and there was a large difference between the *epi* profiles of genotype A compared to genotype D. This phenomenon may explain previous failures during preclinical trials of candidate vaccines against CHB generated thus far, which were designed for broad potency. [68,69,70,71,72,73,74,75]. This may suggest that a genotype- and population-based, multi-epitope vaccine would be the best candidate to combat HBV. Using the cut-off set in this analysis, 21 *epi* from S regions and 6 from PreS2 regions were identified and used to construct tertiary structure of candidate vaccine. The candidate vaccines showed high binding in all alleles except for DRB1*0301. Populations with high coverage of DRB1*0301 have been closely associated with high susceptibility to CHB infection and nonresponse vaccination with envelope proteins [53,76,77].

Among the 3 types of proteins (PreS/S) analyzed in this study, S had the highest binding peptides constituting 79.9% of the sum of all T_epi_ (∑T_epi_), and genotype D had the most epitopes. However, when D_epi_ were compared between genotypes (A and D) and proteins (PreS/S), it was clear that the length of proteins where independent of the *epi* but dependent on the aa compositions and the position of the *epi* within the proteins. While S region was the longest protein (213 aa), it had similar D_epi_ to the smallest protein PreS2_D_ (PreS2 -54 aa). Previous studies that investigated all 7 proteins of HBV and showed that S protein had the most T cell antigenic epitopes [78,79]. However, further investigation should be conducted to determine any relationships between antigenicity and PreS1 protein with least *epi*.

We also compared the frequency of *epi* between 2 genotypes, A and D, and observed that genotype D had generally more immunogenic *epi* than A with exception for PreS2_A_ whose *epi* were 12% more than those of PreS2_D_, (Figure 1). Clinically, these trends correspond to data reported from studies that investigated the prognosis of patients infected with HBV genotype A1 strains compared to genotype D3. Others have reported a 10-fold increased progression to HCC among HBV patients infected with genotype A compared to genotype D and that patients infected with genotype A strains were likely to progress to CHB compared to genotype D [8,80]. Most countries in SSA, including Botswana, have low prevalence of HBV genotype D3 among different risk groups than genotype A1 [25,26,27,81].

Using existing information from studies that investigated the impact of alleles on different HBV outcomes to validate our statistical associations [53,82,83,84,85], we observed that out of 9 alleles, HLA-DRB*1301/2 and *0401 alleles—which had most *epi*—have been associated with spontaneous clearance of HBV infection [18,24,33,68,78,82,85,86,87], and HLA DRB1*0301 that had the least *epi* in our study has been previously associated with susceptibility to HBV infection, autoimmune hepatitis, chronicity, and non-responsiveness to HBV vaccination across different ethnic groups [9,88,89]. This strongly suggests that T_epi_ of *PreS/S* should be explored further as a predictor of the protective effect of HLA class II alleles.

A host immune system can recognize foreign antigens (*epi*) and clear the infection in some cases [90,91,92]; however, most pathogens including HBV can mutate within epitopes, and this may result in an escape from host immune surveillance leading to persistence of infection [93]. This characteristic is regarded as one of the major hindrances in developing high potency therapeutic drugs. This mechanism of aa changes within epitopes (escape mutations) interferes with both peptide processing reducing the intracellular antigen load and downregulation of MHC expression hence increased risk of developing liver malignancies (HCC, LC) among CHB patients [4,5,94,95,96,97,98]. Studies investigating the role of escape mutations within the T cell epitopes are relatively rare. In this study, we observed coordinated aa variations, which reveal genetic dependencies (i.e., *epi* that escaped HLA binding when there were two or more mutations); however, some single aa mutations altered the binding potential. These mutations were termed covariance mutations. For instance, in the proportion of *PreS1* binding peptides from both genotype A and D against alleles shown in Figure 3, 15-mers with core aa sequence ILATVPAVP_84–92_ in PreS1_A_—corresponding to ILQTLPANP_73–81_ in PreS1_D_—weakly binds to alleles: *0101; 0401; 0701; 0802; *1101; 1302; *1501; 5*0101 → PreS1_A_, and *0101; 0401; 0701; 0802; 1302→ PreS1_D_ respectively. The aa *Ala* in genotype A is replaced by aa *Gln* for genotype D [A_(A)_ → Q _(D)_] and [Val_(A)_ → Lue_(D)_]^91^ causing *PreS1*_D_ to pseudo bind to 3 alleles (*1101, 1302, and 5*0101). Additionally, the epitope strongly binds to allele *0802, but the changes in genotype D result in pseudo binding. Table 3, Table 4, Table 5, Table 6 and Table 7 summarize all the core aa of *PreS/S* epitopes that are restricted to 9 HLAs. We observed that the HBV epitope-HLA is greatly influenced by the position of core aa. For instance PreS1_D_
*epi* AFGLGFTPPHGGLLG_51–65_ is a WB to alleles: *0101, *0401 and 5*0101 when using core-aa: FGLGFTPPH_52–60_ but it can only bind to *0701 when the core aa is FTPPHGGLL_57–64_. Furthermore, post *epi* analyses show that there were mutations outside the core aa sequences but had impact on the HBV-HLA blinding. For instance, 2 *epi* with aa residues S: 41–55—FLGGPPVCLGQNSQS and FLGGSPVCLGQNSQS—and all with core aa sequences FLGGPPVCL show different binding affinities thus NB and WB respectively. The escape mutations defined in the present study were those found within the core aa sequence.

Overall, the present computational study facilitates the development of experimental epitope and escape mutation mapping studies.

## 5. Conclusions

Vaccines act by inducing strong immunity to counteract viral antigens presented by MHC-epitopes. However, their success is affected by virus evolution (e.g., escape mutations) within known protective epitopes; hence, multi-epitope, population-based vaccine constructs are preferred in order to generate a potent immunologic response against HBV. We demonstrate the quality of T cell epitopes among different HBV genotypes and reconstructed a candidate multi-epitope population-based vaccine. Our results suggest that among aa variations classified as polymorphisms do exit T-cell escape mutations and. In silico studies should be followed up with preclinical assays to validate the novelty of their findings.

Viral hepatitis (VH) is a global burden, and the WHO has put forth an ambitious goal to eliminate VH as a public threat by 2030. HBV contributes a vast majority (77%) of VH cases and there are no therapeutic cures for chronic hepatitis B infections (CHB). We hypothesized that epitope vaccines are a potential CHB treatment because they can induce strong immune systems with ability to achieve hepatitis B virus surface antigen (HBsAg) loss. While several trials have failed to produce effective vaccines against CHB from T cell epitopes, we aimed to investigate the repertoire of T cell epitopes from different HBV genotypes (A and D), MHC class II alleles with high population coverage in Botswana. In silico analyses were used to map promiscuous epitopes (15-mers) using alleles -9 MHC class II alleles-, and PreS/S sequences -genotype A and D- with high population coverage in Botswana. Some epitopes mapped within PreS/S conserved regions, and none were promiscuous to all alleles suggesting that multi-epitope, population-based vaccines (MEPBV) may be more effective candidate vaccines against CHB compared to previously reported broad potency epitope-based candidate vaccines. Highly promiscuous peptides may also be considered as candidate peptides for designing highly sensitive diagnostic chips since current serological kits may fail to detect other HBV clinical phenotypes. The mapped T epitopes exhibited high mean diversity among genotypes and others had coordinated amino acid variations that were genetically dependent on each other in order to escape epitope-HLA binding.

## Figures and Tables

**Figure 1 viruses-12-00731-f001:**
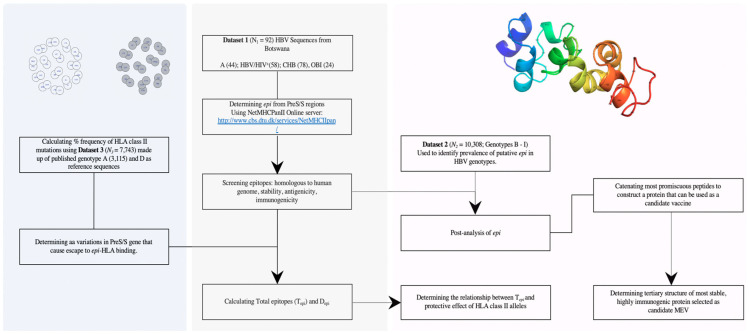
Schema illustrating the flow of data analysis used in this study. N = sample size; SB = strong binding peptides; WB = weak biding peptides; T_epi_ = total predicted epitopes; PreS/S = HBV surface gene; D_epi_ = epitope densities. Sequences were derived from patients with different clinical outcomes: −(HBV/HIV; CHB; OBI)—HIV = human immunodeficiency virus; OBI = occult hepatitis B infection, CHB = chronic hepatitis B infection, HBV/HIV = coinfection. The blue colored segment shows the pipeline used to evaluate the diversity of *epi*. The grey segment is the pipeline used to determine *epi* and measure of promiscuity and conservativeness. The pink segment is the pipeline used to determine the best candidate vaccine.

**Figure 2 viruses-12-00731-f002:**
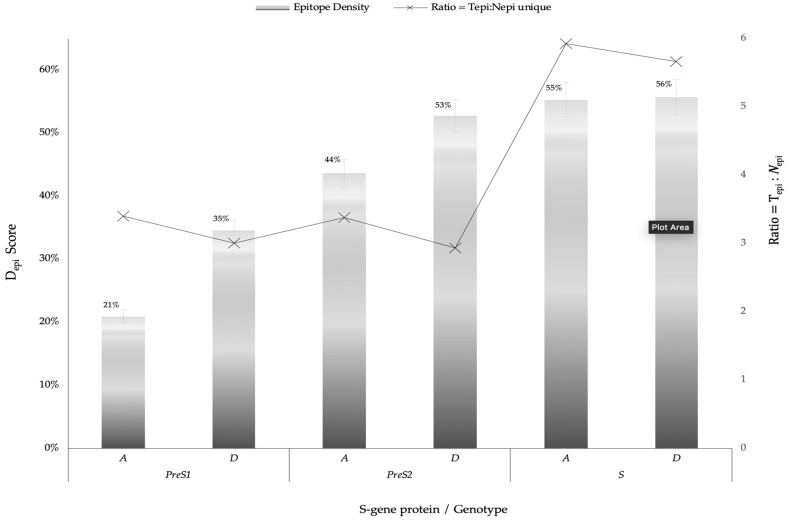
Epitope densities (D_epi_) of different PreS/S proteins stratified by genotype (A or D) and protein (*PreS1*, *PreS2*, *S*). $D_{\mathrm{epi}}=\frac{\;\sum_{i=1}^{n}I\mathrm{epi}}{T_{\mathrm{epi}}\;X\;\mathrm{Protein}\;\mathrm{length}}$ where i can be any protein (PreS/S_A_ versus PreS/S_D_). PreS1_A_ represent genotype A large Hepatitis B surface antigen (HBsAg); PreS1_D_ represent genotype D large HBsAg; PreS2_A_ represent genotype A middle HBsAg; PreS2_D_ represent genotype D middle HBsAg; S_A_ represent genotype A small HBsAg; S_D_ represent genotype D small HBsAg. T_epi_ = Total binding peptides (WB + SB). N_epi_ unique = count of unique binding peptides per each protein.

**Figure 3 viruses-12-00731-f003:**
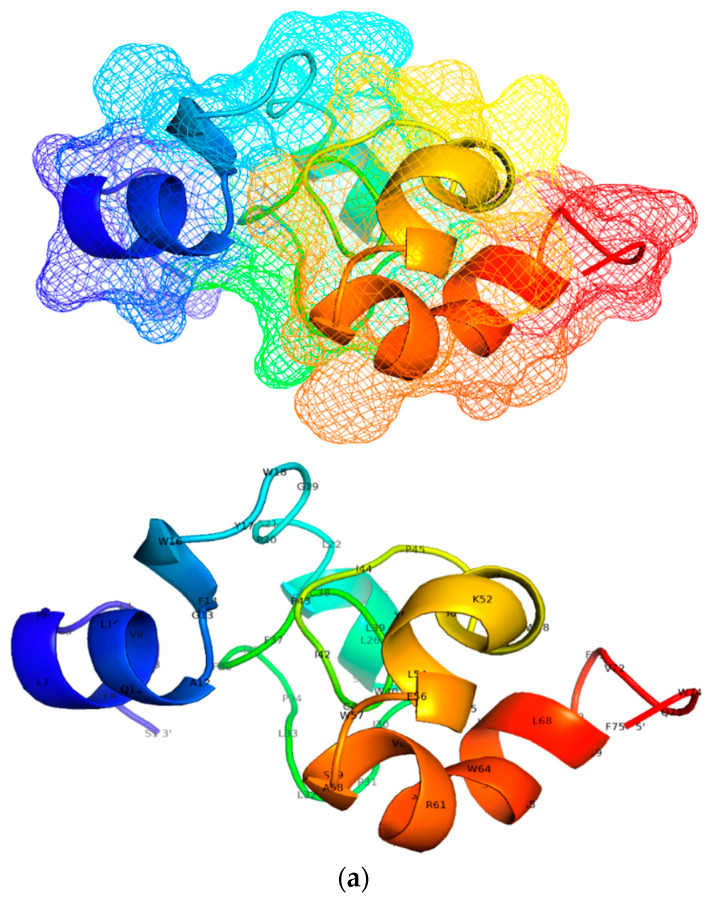
Showing a tertiary structure of candidate vaccines: (**a**) Tertiary structures of candidate epi modelled using 3Dpro webtool. The S_A_ protein in (a) has the aa composition: 5′-SGFLGPLLVLQAGFFWYWGPSLYNILSPFIPLLPIFFCLWCIPIPSSWAFAKYLWEWASVRFSWLSLLVPFVQWF-3′, and had following theoretical properties: antigenicity (0.04), instability index (II = 47.07), estimated half-life in vitro = 1.9 h, molecular weight (mw) = 8878.62, aliphatic index (AI = 114.40) and grand average of hydrophobicity (GRAVY = 0.995), and theoretical alkalinity (pI = 7,76). Using VaxiJen ver2.0 server set at threshold of 0.4, the overall prediction for the protective antigen was 0.53, displaying it as a plausible antigen [66]. (**b**) (5′-MMWYWGPSLYSILSPFLPLLPIFFCLWSGFLGPLLVLQAGFFSWAFGKFLWEWASARFSWLSLLVPFVQWFTCPGYRWMCLRRFIIFLF-3′) protein modelled using from S_D_ epi. The protein had following theoretical properties: antigenicity (0.11), instability index (II = 53), estimated half-life in vitro 30 h, molecular weight (mw) = 10749.99, aliphatic index (AI = 101.91) and grand average of hydrophobicity (GRAVY = 0.965), and theoretical alkalinity (pI = 9.42),). The antigenicity score predicted in both candidate vaccine suggests that they are plausible antigens [66].

**Figure 4 viruses-12-00731-f004:**
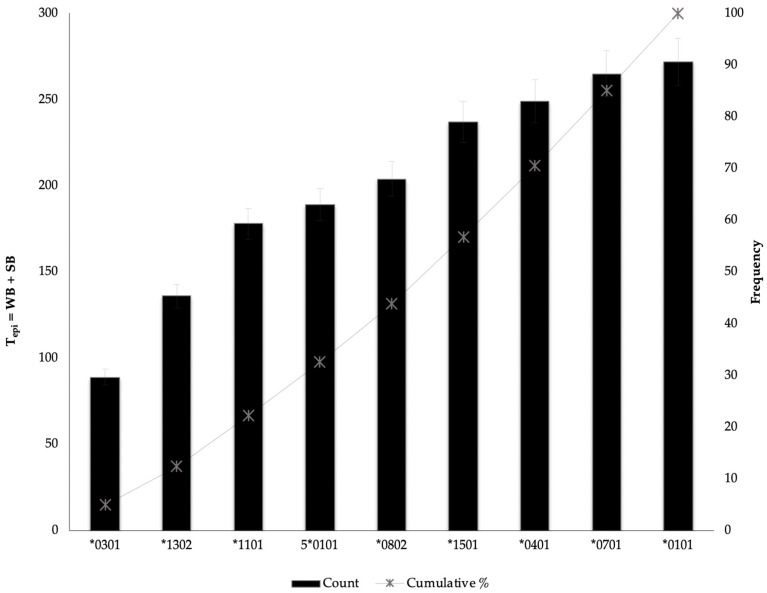
Pareto Analysis applied to rank the T_epi_ of alleles against their percentage frequency.

**Table 1 viruses-12-00731-t001:** Distribution of T cell epitopes restricted to 9 HLA class II alleles with high population coverage in Botswana.

	PreS1		PreS2		S		
	A (120 aa)	D (108 aa)	A (55 aa)	D (55 aa)	A (226 aa)	D (226 aa)	
Total bindings (*n* = 1819) SB (367); WB (1452) (%)	6; 79 (4.7)	7; 107 (6.3)	4; 81 (4.7)	3; 78 (4.5)	122; 619 (40.7)		136; 577 (39.1%)
Unique *epi* (*n* = 297)	25 ^¥^	38 ^¥^	29 ^¥^	24 ^¥^	125 ^¥^		126 ^¥^
Ratio = T_epi_: *N*_epi_	3.4	3.0	3.4	2.9	5.9		5.7
Most active HLA: DRB * (SB; WB)	*0802; *0101	*0401; (*0401, *0101)	*0301; (*0401, *0101)	*0301; 5*0101	*0702; *0401		*0401; *1501
Genotype variation: $\frac{A}{D}$	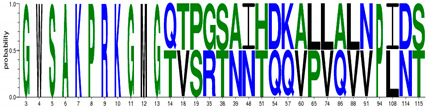		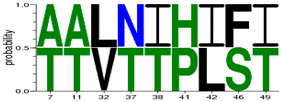		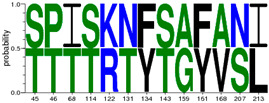		
(A|D *epi*; *p*-value)	A|D > 0.05		A|D > 0.05		A|D > 0.05		

^¥^ indicates existence of *epi* that are common in both genotypes A and D. SB; strong binding peptides. WB; weak binding peptides. aa; amino acids. Web-logo diagrams represent signature aa between consensus sequences of genotype A and D set at a threshold of 100%. HBV; hepatitis B virus. * 0101 means HLA class II allele DRB1*0101 etc. 5*0101 means HLA class II allele DRB5*0101.

**Table 2 viruses-12-00731-t002:** Showing the prevalence of S protein *epi* in other HBV genotypes (B–I).

Epitope Sites in S Protein	AA Sequence	B (*n* = 2905)	C (*n* = 5575)	E (n = 1118)	F (*n* = 477)	G (*n* = 86)	H (*n* = 69)	I (*n* = 78)	Prevalence (%) = 1 − $\left[\frac{Count\;of\;Seq}{10308}\right]$ *%	Degree of Conservation ↓ (+: Variable)	*epi* Previously Discussed
17–31	AFGKFLWEWASARFS	E = 998	C = 32	F = 1					10,0	+	[42]
180–194	AGFFLLTRILTIPQS	B = 62	C = 4627	E = 5	F = 2	G = 75			46,3	++	[43]
90–104	CLIFLLVLLDYQGML	B = 2605	C = 4661	E = 1034	F = 442	G = 84	H = 65	I = 70	86,9	++++	[44]
69–83	CPGYRWMCLRRFIIF	B = 2384	C = 5064	E = 1027	F = 465	G = 84	H = 67	I = 65	88,8	++++	[42]
19–33	FFLLTRILTIPQSLD	B = 63	C = 4717	E = 5	F = 2	G = 76			47,2	++	[45,46,47]
158–172	FGKFLWEWASARFSW	E = 996	C = 31	F = 1					10,0	+	[48]
20–34	FLLTRILTIPQSLDS	B = 63	C = 4709	E = 5	F = 5	G = 78			47,1	++	[49]
93–107	FLLVLLDYQGMLPVC	B = 2616	C = 4720	E = 1031	F = 446	G = 84	H = 66	I = 72	87,7	++++	[50]
161–175	FLWEWASARFSWLSL	B = 1	E = 997	C = 81	F = 3	I = 16			10,7	+	[42]
179–193	FVQWFVGLSPTVWLS	B = 2630	C = 4698	E = 76	G = 85	I = 75			73,4	+++	[51]
18–32	GFFLLTRILTIPQSL	B = 63	C = 4618	E = 5	F = 2	G = 76			46,2	++	-
159–173	GKFLWEWASARFSWL	E = 1002	C = 31	F = 3					10,1	+	-
202–216	GPSLYSILSPFLPLL	C = 19	E = 2						0,2	+	[48]
71–85	GYRWMCLRRFIIFLF	B = 82	C = 5065	E = 1032	F = 464	G = 84	H = 67	I = 65	66,5	+++	-
92–106	IFLLVLLDYQGMLPV	B = 2606	C = 4659	E = 1030	F = 443	G = 84	H = 66	I = 70	86,9	++++	-
195–209	IWMMWYWGPSLYSIL	B = 3	C = 24	E = 2					0,3	+	-
160–174	KFLWEWASARFSWLS	B = 1	E = 1012	C = 34	F = 3	I = 16			10,3	+	-
91–105	LIFLLVLLDYQGMLP	B = 2611	C = 4656	E = 1031	F = 443	G = 84	H = 66	I = 70	86,9	++++	[52]
21–35	LLTRILTIPQSLDSW	B = 63	C = 4754	E = 5	F = 5	G = 77			47,6	++	[53]
94–108	LLVLLDYQGMLPVCP	B = 2653	C = 4729	E = 1032	F = 445	G = 84	H = 66	I = 72	88,1	++++	[54]
15–29	LQAGFFLLTRILTIP	B = 62	C = 4735	E = 5	F = 2	G = 76			47,3	++	[54]
192–206	LSVIWMMWYWGPSLY	B = 772	C = 4249	E = 889	G = 1	I = 50			57,8	++	[44]
95–109	LVLLDYQGMLPVCPL	B = 2619	C = 4708	E = 1022	F = 444	G = 84	H = 66	I = 73	87,5	++++	[55,56]
162–176	LWEWASARFSWLSLL	B = 7	E = 1001	C = 88	F = 445	G = 2	H = 65	I = 75	16,3	+	[54]
205–219	LYSILSPFLPLLPIF	C = 21	E = 2						0,2	+	-
178–192	PFVQWFVGLSPTVWL	B = 2649	C = 4723	E = 76	G = 85	I = 74			73,8	+++	-
70–84	PGYRWMCLRRFIIFL	B = 2375	C = 5117	E = 1033	F = 466	G = 85	H = 67	I = 65	89,3	++++	[49]
66–80	PPTCPGYRWMCLRRF	B = 60	C = 751	E = 14	F = 464	G = 77	H = 68		13,9	+	[44]
203–217	PSLYSILSPFLPLLP	C = 19	E = 2						0,2	+	-
67–81	PTCPGYRWMCLRRFI	B = 60	C = 750	E = 14	F = 464	G = 77	H = 68		13,9	+	[57]
16–30	QAGFFLLTRILTIPQ	B = 62	C = 4645	E = 5	F = 2	G = 76			46,5	++	[58]
181–195	QWFVGLSPTVWLSVI	B = 2588	C = 4310	E = 73	G = 6	I = 71			68,4	+++	[58]
38–52	SLNFLGGTTVCLGQN	B = 5	C = 2	E = 2					0,1	+	[55,56,57]
204–218	SLYSILSPFLPLLPI	C = 19	E = 2						0,2	+	[44]
193–207	SVIWMMWYWGPSLYS	B = 3	C = 24	E = 2					0,3	+	-
155–169	SWAFGKFLWEWASAR	E = 998	C = 31	F = 1					10,0	+	-
68–82	TCPGYRWMCLRRFII	B = 63	C = 757	E = 17	F = 466	G = 77	H = 68		14,0	+	-
37–51	TSLNFLGGTTVCLGQ	B = 5	C = 2	E = 2					0,1	+	[59,60]
194–208	VIWMMWYWGPSLYSI	B = 3	C = 24	E = 2					0,3	+	-
14–28	VLQAGFFLLTRILTI	B = 61	C = 4676	E = 5	F = 2	G = 76			46,8	++	[61]*
180–194	VQWFVGLSPTVWLSV	B = 2636	C = 4584	E = 77	G = 6	I = 75			71,6	+++	[62]
156–170	WAFGKFLWEWASARF	E = 1000	C = 32	F = 1					10,0	+	[44]
163–177	WEWASARFSWLSLLV	B = 7	E = 1002	C = 87	F = 445	G = 2	H = 64	I = 75	16,3	+	-
182–196	WFVGLSPTVWLSVIW	B = 2506	C = 4125	E = 73	G = 6	I = 71			65,8	+++	[63]
201–215	WGPSLYSILSPFLPL	C = 19	E = 2						0,2	+	[52]
196–210	WMMWYWGPSLYSILS	B = 3	C = 25	E = 2					0,3	+	-
36–50	WTSLNFLGGTTVCLG	B = 5	C = 2	E = 2					0,1	+	-
35–49	WWTSLNFLGGTTVCL	B = 5	C = 2	E = 2					0,1	+	-
199–213	WYWGPSLYSILSPFL	C = 18	E = 2						0,2	+	-
206–220	YSILSPFLPLLPIFF	C = 19	E = 2						0,2	+	-
200–214	YWGPSLYSILSPFLP	C = 18	E = 2						0,2	+	-

+ The degree of conservation. The scale used: if score > = 100, then highly conserved and will be denoted by ‘+++++’. elif score > = 85: then semi conserved = ‘++++’. elif score > = 60: region of mutation and is denoted by ‘+++’. elif score > = 20, then highly variable region = ‘++’. else: high escape mutation = ‘+’. n represents the number of sequences used in the analysis. B represents full-length genotype B sequences, C represents full-length genotype C sequences, etc. 17–31 representing the beginning the position occupied by the 14-mer epitope predicts (e.g., 17 is the first amino acid of the epitope, while 31 is the last amino acid of the epitope).

**Table 3 viruses-12-00731-t003:** List of most promiscuous SB *epi* mapped from S protein.

S: Residues	AA Sequence	Geno	Core aa	HLA Class II Alleles								*S: Residues*	AA Sequence	Geno	Core aa	HLA Class II Alleles								
69–83	CPGYRWMCLRRFIIF	A	YRWMCLRRF	*0101	*0301	*0401	*0701	*0802	*1302			68–82	TCPGYRWMCLRRFII	D	YRWMCLRRF	*0101	*0301	*0401	*0701	*0802		*1302	*1501	
	CPGYRWMCLRRFIIF	D										168–182	VRFSWLSLLVPFVQW	A	WLSLLVPFV	*0101		*0401	*0701	*0802	*1101	*1302	*1501	5*0101
*68–82*	ICPGYRWMCLRRFII	A										14–28	VLQAGFFLLTRILTI	A	FLLTRILTI		*0301	*0401		*0802		*1302	*1501	5*0101
*70–84*	PGYRWMCLRRFIIFL	D											VLQAGFFLLTRILTI	D										
	PGYRWMCLRRFIIFL	A		*0101	*0301					*1501		165–179	WASARFSWLSLLVPF	D	FSWLSLLVP	*0101		*0401		*0802	*1101			5*0101
*71–85*	GYRWMCLRRFIIFLF	A										174–188	SLLVPFVQWFVGLSP	A	FVQWFVGLS	*0101				*0802	*1101		*1501	
	GYRWMCLRRFIIFLF	D											SLLVPFVQWFVGLSP	D										
*67–85*	PICPGYRWMCLRRFI	A										6–20	SGFLGPLLVLQAGFF	A	LGPLLVLQA	*0101			*0701	*0802	*1101		*1501	5*0101
*66–84*	PPICPGYRWMCLRRF	A		*0101						*1501	5*0101		SGFLGPLLVLQAGFF	D					*0701					
*67–81*	PTCPGYRWMCLRRFI	D								*1501		7–21	GFLGPLLVLQAGFFL	D										
*147–161*	CTCIPIPSSWAFAKY	A	CIPIPSSWA	*0101		*0401	*0701	*0802	*1501				GFLGPLLVLQAGFFL	A										
	CTCIPIPSSWAFGKF	D										5–19	TSGFLGPLLVLQAGF	D										
*145–159*	GNCTCIPIPSSWAFA	A											TSGFLGPLLVLQAGF	A										
*146–160*	NCTCIPIPSSWAFAK	A										93–107	FLLVLLDYQGMLPVC	D	LVLLDYQGM	*0101		*0401		*0802	*1101		*1501	
	NCTCIPIPSSWAFGK	D									5*0101		FLLVLLDYQGMLPVC	A						*0802	*1101			
*177–191*	VPFVQWFVGLSPTVW	D	WFVGLSPTV	*0101		*0401	*0701	*0802	*1101		5*0101	92–106	IFLLVLLDYQGMLPV	A						*0802	*1101			
	VPFVQWFVGLSPTVW	A											IFLLVLLDYQGMLPV	D						*0802	*1101			
*168–182*	ARFSWLSLLVPFVQW	D	WLSLLVPFV	*0101		*0401	*0701	*0802	*1101	*1501	5*0101	182–196	WFVGLSPTVWLSVIW	D	VGLSPTVWL		*0301	*0401		*0802	*1302		*1501	
*169–183*	RFSWLSLLVPFVQWF	D											WFVGLSPTVWLSVIW	A			*0301	*0401		*0802	*1302			
	RFSWLSLLVPFVQWF	A										97–111	LLDYQGMLPVCPLIP	A	YQGMLPVCP	*0101		*0401	*0701		*1101		*1501	5*0101
*167–181*	SARFSWLSLLVPFVQ	D											LLDYQGMLPVCPLIP	D				*0401	*0701		*1101			
*176–189*	LVPFVQWFVGLSPTV	A	WFVGLSPTV	*0101		*0401	*0701	*0802			5*0101	96–110	VLLDYQGMLPVCPLI	A				*0401	*0701		*1101			
	LVPFVQWFVGLSPTV	D											VLLDYQGMLPVCPLI	D				*0401	*0701		*1101			
*166–180*	ASARFSWLSLLVPFV	D		*0101		*0401	*0701	*0802			5*0101	72–86	YRWMCLRRFIIFLFI	A	WMCLRRFII				*0701	*0802	*1101		*1501	5*0101
*170–184*	FSWLSLLVPFVQWFV	A							*1101				YRWMCLRRFIIFLFI	D	WMCLRRFII									
	FSWLSLLVPFVQWFV	D										194–208	VIWMMWYWGPSLYNI	A	WYWGPSLYN	*0101		*0401	*0701	*0802	*1101			5*0101
*167–181*	SVRFSWLSLLVPFVQ	A									5*0101													

PreS1_A_ represent genotype A *epi* derived from sequences of large Hepatitis B surface antigen (HBsAg); PreS1_D_ represent genotype D *epi* derived from sequences of large HBsAg; PreS2_A_ represent genotype A *epi* derived from sequences of middle HBsAg; PreS2_D_ represent genotype D *epi* derived from sequences of middle HBsAg; S_A_ represent genotype A *epi* derived from sequences of small HBsAg; S_D_ represent genotype D *epi* derived from sequences of small HBsAg. *0101 means HLA class II allele DRB1*0101 e.tc. 5*0101 means HLA class II allele DRB5*0101.

**Table 4 viruses-12-00731-t004:** Highlighting most promiscuous T cell epitopes restricted to 9 HLA class II alleles.

Epitope Site	AA Sequence	HBV Protein	Core AA	HLA Class II Alleles	Previously Discussed *epi*
34–48	PVPNIASHISSISSR	PreS2_A_	IASHISSIS	1*0101, 1*0401, 1*0701, 1*0802, 1*1101, 1*1302, 1*1501, 5*0101	-
39–53	ASHISSISSRTGDPA	PreS2_A_	ISSISSRTG	1*0101, 1*0401, 1*0701, 1*0802, 1*1101, 1*1501, 5*0101	-
38–52	IASHISSISSRTGDP	PreS2_A_	ISSISSRTG	1*0101, 1*0401, 1*0701, 1*0802, 1*1101, 1*1501, 5*0101	-
37–51	TTASPLSSIFSRIGD	PreS2_D_	LSSIFSRIG	1*0101, 1*0401, 1*0701, 1*0802, 1*1101, 1*1501, 5*0101	-
38–52	TASPLSSIFSRIGDP	PreS2_D_	LSSIFSRIG	1*0101, 1*0401, 1*0701, 1*0802, 1*1101, 1*1501, 5*0101	-
39–53	ASPLSSIFSRIGDPA	PreS2_D_	LSSIFSRIG	1*0101, 1*0401, 1*0701, 1*0802, 1*1101, 1*1501, 5*0101	-
208–222	ILSPFIPLLPIFFCL	S_A_	FIPLLPIFF	1*0101, 1*0401, 1*0701, 1*0802, 1*1101, 1*1501, 5*0101	-
209–223	LSPFIPLLPIFFCLW	S_A_	FIPLLPIFF	1*0101, 1*0401, 1*0701, 1*0802, 1*1101, 1*1501, 5*0101	-
207–221	NILSPFIPLLPIFFC	S_A_	FIPLLPIFF	1*0101, 1*0401, 1*0701, 1*0802, 1*1101, 1*1302, 5*0101	-
149–163	CIPIPSSWAFAKYLW	S_A_	IPSSWAFAK	1*0101, 1*0401, 1*0701, 1*0802, 1*1101, 1*1501, 5*0101	-
6–20	SGFLGPLLVLQAGFF	S_A_	LGPLLVLQA	1*0101, 1*0401, 1*0701, 1*0802, 1*1101, 1*1501, 5*0101	-
200–214	YWGPSLYNILSPFIP	S_A_	LYNILSPFI	1*0101, 1*0401, 1*0701, 1*0802, 1*1101, 1*1302, 1*1501	-
199–213	WYWGPSLYNILSPFI	S_A_	LYNILSPFI	1*0101, 1*0401, 1*0701, 1*0802, 1*1101, 1*1501, 5*0101	-
164–178	EWASVRFSWLSLLVP	S_A_	VRFSWLSLL	1*0101, 1*0401, 1*0802, 1*1101, 1*1302, 1*1501, 5*0101	[64]
168–182	VRFSWLSLLVPFVQW	S_A_	WLSLLVPFV	1*0101, 1*0401, 1*0701, 1*0802, 1*1101, 1*1302, 1*1501, 5*0101	[44]
169–183	RFSWLSLLVPFVQWF	S_A_	WLSLLVPFV	1*0101, 1*0401, 1*0701, 1*0802, 1*1101, 1*1501, 5*0101	[65]
156–170	WAFAKYLWEWASVRF	S_A_	YLWEWASVR	1*0101, 1*0301, 1*0401, 1*0701, 1*0802, 1*1101, 1*1501, 5*0101	[49]
208–222	ILSPFLPLLPIFFCL	S_D_	FLPLLPIFF	1*0101, 1*0401, 1*0701, 1*0802, 1*1101, 1*1501, 5*0101	-
209–223	LSPFLPLLPIFFCLW	S_D_	FLPLLPIFF	1*0101, 1*0401, 1*0701, 1*0802, 1*1101, 1*1501, 5*0101	-
155–169	SWAFGKFLWEWASAR	S_D_	FLWEWASAR	1*0101, 1*0301, 1*0401, 1*0701, 1*0802, 1*1101, 1*1501	-
6–20	SGFLGPLLVLQAGFF	S_D_	LGPLLVLQA	1*0101, 1*0401, 1*0701, 1*0802, 1*1101, 1*1501, 5*0101	[44]
199–212	WYWGPSLYSILSPFL	S_D_	LYSILSPFL	1*0101, 1*0401, 1*0802, 1*1101, 1*1302, 1*1501, 5*0101	-
167–181	SARFSWLSLLVPFVQ	S_D_	WLSLLVPFV	1*0101, 1*0401, 1*0701, 1*0802, 1*1101, 1*1501, 5*0101	-
168–182	ARFSWLSLLVPFVQW	S_D_	WLSLLVPFV	1*0101, 1*0401, 1*0701, 1*0802, 1*1101, 1*1501, 5*0101	[44]
169–183	RFSWLSLLVPFVQWF	S_D_	WLSLLVPFV	1*0101, 1*0401, 1*0701, 1*0802, 1*1101, 1*1501, 5*0101	[42,49]
197–211	MMWYWGPSLYSILSP	S_D_	WYWGPSLYS	1*0101, 1*0401, 1*0701, 1*0802, 1*1101, 1*1501, 5*0101	-
68–82	TCPGYRWMCLRRFII	S_D_	YRWMCLRRF	1*0101, 1*0301, 1*0401, 1*0701, 1*0802, 1*1101, 1*1501, 5*0101	-

*0101 means HLA class II allele DRB1*0101 e.tc. 5*0101 means HLA class II allele DRB5*0101.

**Table 5 viruses-12-00731-t005:** The prevalence of S gene escape mutations.

Protein	AA Sequence	S Protein AA Residues (*epi*)	Escape Mutations	Count in A (*n* = 3115)	Count in D (*n* = 4628)	Count in Other Genotypes
S_A_	MENITSGFLGPQLV	1–14	L12Q	13	1	3
		1–14	I4T	13	25	91*^∆^*
		1–14	I4Stop	10	-	-
S_A_	GPLLVLQAGFFLLTR	10–24	L15Stop	2	-	17
S_A_	LNFPGGSPVCLGQNS	39–53	L42P	13	6	53
S_A_	TRILTIPQ *LDSWWT	23 -37	S31Stop	6	12	-
S_A_	DLWWTSLNFLGDPPV	33–47	G44D	4	-	16
	DSWWTSLNFLGESPV		G44E	105	87	826
	IPIPSSWGFAKYLWE	150–164	A157G	10	5	3
S_A_	IPIPSSWAFVKYLWE		A159V	20	3	234
S_A_	PPICPGYRWMCQRRF	66–80	L77R	7	2	43
			L77Q	6	-	18
S_A_	PPICPGYRWMCLR *F		R79Stop	9	10	20
			R79H	13	58	49
S_A_	LIFLLVLLDYQDMLP	91–105	G102D	1	2	3
S_A_	CLIFLLVLLDYQGML	90–104	D99Stop	15	5	8
			Y100C	196	12	17
			M103I	31	31	29
S_A_	SLLVPFVQWFVGLTP LLVPFVQWFEGLSPT	174–188	S187T	1	-	-
			G185E	6	15	19
			V184E	1	2	2
			L186P	10	-	5
S_A_	SPTVWLLAIWMMWYW	187–193	S193L	104	257	612
			A194V	494	*^∆ *^*	
S_A_	GPSLYNISSPFIPLL	202–216	L209S	4	9	12
S_D_	ENITSGCLGPLLVLQAGF	2–16	F8C	-	1	-
S_A_	YLWEWASVRFSWPSL	161–175	L173P	1	3	11
S_A_	SPFIPLLLIFFCLWV	210–224	P214L	11	41	23
S_D_	SSWAFGKFLWEWASA	154–168 208–222	K160N	1	7	9
S_D_	ILSPYILLLPIFFCI		F212Y + L213I + P214L	-; -; 11	40; 202; 41	3; ^∆ *^;23
	NITSGFLGLLLVLQA	4 -18	P11L	2	4	3
S_D_	MENITSGFLGPLLVL	1–15	T5P; N3S + I4T + T5A	2; -	4; 2	4; -29
S_D_	SWWTSLNFLGETTVC	34–48	G44E	105	87	826
S_D_	SWWTSLNFRGGTTVC		L42R	14	8	14
S_D_	LSVIWMMWYWGPNLY	192–206	S204N	136	194	848
**List of aa Variations within Genotype A and D *epi***	M1E, M1L, M1K, M1I, M1V, M1T, M1R, E2H, E2Q, E2*, E2V, E2K, E2A, E2G, E2D, N3H, N3E, N3P, N3A, N3C, N3D, N3I, N3R, N3Y, N3K, N3T, N3G, N3S, I4L, I4R, I4Y, I4Q, I4F, I4M, I4H, I4K, I4A, I4P, I4S, I4N, I4V, I4T, T5Y, T5Q, T5L, T5R, T5E, T5K, T5V, T5P, T5I, T5S, T5A, S6G, S6F, S6A, S6P, S6T, S6L, G7C, G7D, G7*, G7A, G7V, G7K, G7E, G7R, F8C, F8A, F8G, F8V, F8Y, F8I, F8H, F8P, F8S, F8L, L9R, L9K, L9V, L9I, L9H, L9Q, L9P, G10D, G10*, G10V, G10Q, G10T, G10A, G10K, G10E, G10R, P11A, P11T, P11H, P11L, L12V, L12E, L12M, L12R, L12P, L12Q, L13Q, L13V, L13I, L13F, L13R, L13H, L13P, V14R, V14L, V14E, V14I, V14M, V14G, V14A, L15E, L15K, L15T, L15*, L15I, L15F, L15V, L15S, Q16K, Q16E, Q16H, Q16L, Q16R, Q16P, A17R, A17T, A17S, A17P, A17V, A17G, A17E, G18W, G18K, G18E, G18A, G18R, G18V, F19I, F19V, F19L, F19S, F19Y, F19C, F20I, F20Y, F20L, F20S, L21V, L21M, L21F, L21*, L21W, L21S, L22Q, L22M, L22V, L22F, L22S, L22*, L22W, T23F, T23R, T23Q, T23S, T23P, T23A, T23I, R24N, R24T, R24Q, R24E, R24I, R24G, R24S, R24K, I25R, I25S, I25F, I25N, I25A, I25T, I25V, L26Y, L26I, L26F, L26Q, L26P, L26H, L26R, T27R, T27P, T27S, T27A, T27K, T27I, I28K, I28L, I28V, I28T, I28M, P29E, P29A, P29F, P29S, P29Q, P29T, P29L, Q30M, Q30S, Q30P, Q30A, Q30L, Q30H, Q30R, Q30K, S31K, S31I, S31D, S31T, S31C, S31G, S31R, S31N, L32V, L32G, L32Q, L32R, L32I, L32P, D33V, D33H, D33Y, D33E, D33N, D33G, S34*, S34A, S34T, S34W, S34P, S34L, W35P, W35C, W35*, W35G, W35R, W35L, W36E, W36G, W36R, W36*, W36L, T37D, T37P, T37L, T37I, T37S, T37N, T37A, S38E, S38A, S38Y, S38F, S38P, L39I, L39V, L39R, L39H, L39F, L39P, N40Q, N40H, N40K, N40I, N40D, N40S, F41Q, F41I, F41Y, F41C, F41L, F41S, L42*, L42S, L42I, L42V, L42Q, L42R, L42P, G43V, G43W, G43K, G43R, G43E, G44Q, G44K, G44R, G44A, G44D, G44V, G44E, S45M, S45D, S45G, S45E, S45H, S45Q, S45R, S45I, S45K, S45N, S45V, S45L, S45P, S45A, S45T, P46N, P46R, P46S, P46I, P46A, P46H, P46L, P46T, V47Q, V47N, V47L, V47P, V47M, V47E, V47R, V47K, V47A, V47G, V47T, C48W, C48L, C48F, C48R, C48G, C48S, C48Y, L49A, L49T, L49C, L49S, L49I, L49V, L49F, L49H, L49R, L49P, G50C, G50W, G50P, G50R, G50D, G50V, G50S, G50A, Q51E, Q51K, Q51H, Q51P, Q51R, Q51L, N52T, N52G, N52H, N52K, N52I, N52D, N52S, S53G, S53M, S53*, S53T, S53P, S53W, S53L, Q54K, Q54*, Q54H, Q54L, Q54P, Q54R, S55T, S55Y, S55A, S55P, S55F, S55C, P56A, P56*, P56S, P56R, P56H, P56L, P56Q, T57S, T57A, T57I, S58Y, S58A, S58L, S58T, S58P, S58F, S58C, N59L, N59I, N59R, N59H, N59D, N59K, N59S, H60D, H60S, H60N, H60Y, H60Q, H60R, H60P, S61T, S61P, S61*, S61L, P62S, P62Q, P62L, T63V, T63F, T63N, T63S, T63A, T63I, S64W, S64P, S64F, S64Y, S64C, C65G, C65F, C65S, C65Y, C65R, P66S, P66L, P66T, P66A, P66Q, P66H, P67T, P67A, P67R, P67S, P67L, P67Q, I68D, I68P, I68V, I68F, I68S, I68N, I68A, I68T, C69F, C69G, C69S, C69R, C69Y, C69W, C69*, P70S, P70H, P70R, P70A, P70T, P70L, G71S, G71D, G71W, G71V, Y72S, Y72D, Y72N, Y72H, Y72F, Y72C, R73L, R73C, R73P, R73H, W74C, W74G, W74R, W74*, W74S, W74L, M75K, M75L, M75R, M75S, M75V, M75T, M75I, C76G, C76*, C76R, C76W, C76S, C76F, C76Y, L77P, L77G, L77M, L77V, L77Q, L77R, R78I, R78G, R78W, R78P, R78L, R78Q, R79P, R79G, R79L, R79C, R79S, R79H, F80G, F80N, F80Y, F80I, F80L, F80S, I81S, I81Y, I81N, I81F, I81M, I81V, I81T, I82F, I82N, I82V, I82M, I82T, I82L, F83Y, F83I, F83L, F83C, F83S, L84F, L84H, L84P, F85Q, F85R, F85P, F85L, F85Y, F85S, F85C, I86M, I86K, I86L, I86F, I86V, I86T, L87M, L87V, L87R, L87Q, L87P, L88R, L88V, L88M, L88Q, L88P, L89V, L89Y, L89T, L89R, L89I, L89Q, L89P, C90G, C90W, C90N, C90I, C90*, C90Y, C90F, C90S, L91G, L91F, L91P, L91R, L91H, I92H, I92N, I92V, I92M, I92L, I92S, I92T, F93R, F93W, F93A, F93Y, F93I, F93L, F93C, F93S, L94G, L94*, L94V, L94F, L94M, L94W, L94S, L95P, L95F, L95M, L95V, L95S, L95*, L95W, V96I, V96E, V96N, V96F, V96L, V96C, V96D, V96A, V96G, L97C, L97R, L97V, L97H, L97F, L97P, V98M, V98W, V98A, V98Q, V98P, V98R, V98L, D99H, D99K, D99V, D99A, D99E, D99N, D99G, Y100P, Y100*, Y100K, Y100D, Y100H, Y100N, Y100L, Y100W, Y100S, Y100F, Y100C, Q101P, Q101L, Q101*, Q101K, Q101H, Q101R, G102N, G102R, G102A, G102D, G102C, G102S, G102V, M103K, M103R, M103L, M103T, M103V, M103I, L104Q, L104M, L104G, L104V, L104W, L104S, L104F, P105T, P105S, P105H, P105A, P105L, P105R, V106L, V106R, V106Q, V106D, V106E, V106G, V106I, V106A, C107A, C107W, C107*, C107G, C107S, C107Y, C107R, P108A, P108T, P108S, P108L, P108H, L109R, L109V, L109I, L109Q, L109M, L109P, I110R, I110H, I110V, I110N, I110F, I110S, I110P, I110T, I110M, I110L, A111N, A111R, A111T, A111Q, A111S, A111L, A111P, G112V, G112K, G112N, G112E, G112A, G112R, S113C, S113R, S113F, S113L, S113Y, S113K, S113P, S113N, S113A, S113T, T114V, T114*, T114L, T114R, T114N, T114M, T114K, T114A, T114P, T114S, T115P, T115S, T115A, T115I, T115N, T116S, T116P, T116A, T116I, T116N, S117Y, S117A, S117V, S117R, S117I, S117G, S117N, S117T, T118W, T118N, T118G, T118I, T118E, T118P, T118S, T118K, T118R, T118M, T118A, T118V, G119S, G119W, G119T, G119*, G119C, G119V, G119E, G119R, P120H, P120N, P120R, P120L, P120A, P120Q, P120S, P120T, C121T, C121G, C121F, C121L, C121Y, C121S, C121W, C121R, K122G, K122V, K122E, K122N, K122T, K122I, K122S, K122Q, K122R, T123G, T123V, T123P, T123S, T123N, T123I, T123A, C124G, C124R, C124W, C124Y, C124S, C124F, T125P, T125I, T125A, T125M, T126P, T126Q, T126K, T126G, T126M, T126L, T126V, T126N, T126A, T126S, T126I, P127F, P127R, P127V, P127H, P127S, P127A, P127I, P127T, P127L, A128T, A128R, A128G, A128V, Q129E, Q129K, Q129L, Q129P, Q129N, Q129H, Q129R, G130V, G130*, G130D, G130K, G130A, G130E, G130S, G130R, G130N, N131Q, N131G, N131D, N131H, N131K, N131S, N131I, N131A, N131P, N131T, S132H, S132Y, S132P, S132C, S132F, M133R, M133G, M133K, M133V, M133Q, M133S, M133I, M133L, M133T, F134K, F134D, F134W, F134T, F134Q, F134C, F134R, F134V, F134H, F134S, F134L, F134N, F134I, F134Y, P135S, P135R, P135T, P135H, P135L, S136T, S136C, S136A, S136L, S136P, S136F, S136Y, C137S, C137R, C137*, C137Y, C137W, C138S, C138*, C138W, C138G, C138R, C138Y, C139S, C139G, C139*, C139Y, C139W, C139R, T140P, T140K, T140M, T140A, T140L, T140I, T140S, K141N, K141Q, K141T, K141I, K141R, K141E, P142T, P142A, P142H, P142R, P142I, P142S, P142L, T143W, T143F, T143A, T143P, T143M, T143L, T143S, D144Y, D144N, D144V, D144A, D144G, D144E, G145D, G145V, G145I, G145S, G145K, G145E, G145A, G145R, N146I, N146T, N146K, N146S, N146D, C147W, C147S, C147R, C147G, C147Y, T148S, T148I, T148A, C149S, C149G, C149W, C149Y, C149R, I150S, I150N, I150M, I150V, I150T, P151S, P151R, P151T, P151L, P151H, I152H, I152F, I152L, I152T, I152V, P153A, P153T, P153L, P153Q, P153S, S154Q, S154A, S154H, S154V, S154T, S154P, S154L, S155T, S155A, S155Y, S155F, S155P, W156S, W156C, W156G, W156R, W156*, W156L, A157S, A157N, A157D, A157P, A157V, A157T, A157G, F158I, F158Y, F158S, F158L, A159P, A159Q, A159L, A159S, A159T, A159R, A159E, A159V, A159G, K160Q, K160T, K160E, K160G, K160S, K160N, K160R, Y161K, Y161C, Y161V, Y161H, Y161I, Y161L, Y161S, Y161F, L162S, L162*, L162R, L162V, L162I, L162P, L162Q, W163G, W163C, W163R, E164*, E164K, E164A, E164V, E164D, E164G, W165Q, W165*, W165G, W165C, W165S, W165R, W165L, A166S, A166D, A166P, A166T, A166V, A166G, S167Y, S167E, S167T, S167G, S167P, S167L, V168L, V168I, V168F, V168G, V168D, V168S, V16887T, V168P, V168A, R169G, R169L, R169S, R169C, R169P, R169H, F170V, F170L, F170S, S171N, S171Y, S171C, S171L, S171P, S171F, W172M, W172F, W172G, W172R, W172S, W172L, W172C, W172*, P173I, P173Y, P173S, P173V, P173F, P173L, S174G, S174K, S174C, S174R, S174T, S174I, S174N, L175I, L175*, L175F, L175S, L176I, L176V, L176Q, L176P, V177P, V177I, V177S, V177G, V177E, V177M, V177L, V177A, P178G, P178R, P178H, P178A, P178L, P178S, P178Q, F179I, F179C, F179L, F179Y, F179S, V180P, V180L, V180E, V180F, V180I, V180A, Q181K, Q181E, Q181L, Q181H, Q181R, W182F, W182G, W182S, W182R, W182C, W182L, W182*, F183V, F183Y, F183I, F183L, F183S, F183C, V184L, V184P, V184D, V184F, V184I, V184E, V184G, V184A, G185K, G185L, G185W, G185R, G185A, G185E, L186M, L186A, L186R, L186C, L186S, L186I, L186V, L186F, L186P, L186H, S187A, S187T, S187C, S187P, S187F, S187L, P188A, P188S, P188R, P188T, P188H, P188L, T189L, T189N, T189A, T189S, T189P, T189I, V190S, V190E, V190P, V190D, V190G, V190I, V190F, V190A, W191K, W191S, W191*, W191G, W191L, W191R, W191C, L192R, L192H, L192V, L192F, L192P, S193*, S193T, S193P, S193F, S193L, A194G, A194D, A194S, A194T, A194F, A194L, A194I, A194V, I195K, I195V, I195L, I195T, I195M, W196G, W196V, W196K, W196R, W196*, W196S, W196L, M197L, M197K, M197R, M197V, M197I, M197T, M198L, M198V, M198R, M198K, M198T, M198I, W199P, W199C, W199R, W199G, W199S, W199*, W199L, Y200H, Y200*, Y200L, Y200N, Y200S, Y200C, Y200F, W201C, W201L, W201G, W201S, W201R, W201*, G202V, G202E, G202R, G202A, P203H, P203T, P203G, P203S, P203A, P203L, P203Q, P203R, S204Q, S204H, S204C, S204D, S204I, S204T, S204G, S204K, S204R, S204N, L205Q, L205R, L205V, L205P, L205M, Y206G, Y206P, Y206Q, Y206*, Y206I, Y206W, Y206V, Y206D, Y206R, Y206L, Y206N, Y206S, Y206F, Y206H, Y206C, N207G, N207K, N207C, N207D, N207P, N207H, N207I, N207T, N207R, N207S, I208L, I208C, I208F, I208S, I208V, I208N, I208T, L209K, L209A, L209F, L209*, L209G, L209M, L209S, L209W, L209V, S210M, S210G, S210I, S210T, S210K, S210R, S210N, P211A, P211T, P211S, P211R, P211L, P211H, F212C, F212S, F212L, F212Y, I213A, I213V, I213S, I213F, I213T, I213M, I213L, P214G, P214R, P214T, P214H, P214S, P214Q, P214L, L215D, L215I, L215V, L215T, L215M, L215Q, L215R, L215P, L216T, L216G, L216C, L216R, L216Y, L216V, L216I, L216S, L216F, L216*, P217A, P217T, P217Q, P217S, P217L, I218P, I218K, I218S, I218F, I218N, I218T, I218L, F219C, F219I, F219L, F219S, F220V, F220I, F220S, F220W, F220Y, F220L, F220C, C221V, C221R, C221W, C221L, C221S, C221G, C221F, C221Y, H222Q, H222V, H222S, H222R, H222F, H222I, H222P, H222L, W223F, W223G, W223C, W223R, W223*, W223L, V224L, V224I, V224E, V224G, V224A, Y225A, Y225*, Y225N, Y225D, Y225L, Y225I, Y225H, Y225C, Y225F, Y225S, I226R, I226H, I226L, I226F, I226N, I226V, I226T, I226M, I226S,					

1R represents the position of the mutation ‘1’ and R is the change in amino acid “candidate escape mutations”. ^∆^ Genotype C sequences were excluded in the analysis. ^∆*^ genotype B, C, D, E, F, G, H and I sequences were excluded in this analysis. * = Stop codon.

**Table 6 viruses-12-00731-t006:** Summary of *PreS1* epitopes binding to respective alleles, and mutations that exists in the Botswana HBV sequences.

Epitope Position	Core AA Sequence	HLA_ DRB*1/5_ Genotype A	Epitopes Position	Core AA Sequence	HLA_ DRB*1/5_ Genotype D	Mutations Relative to Protein: (WT, aa#, Mut)	
						A	D
84–92	ILATVPAVP ^¥^	*0802	73–81	ILQTLPANP		I84V, I84T, A86V, A86T, V88M, V88L, A90P, A90V, P92L	I73M, L75H, L77M, L77V, A79E
85–93	LATVPAVPP ^¥^	*˜0802	74–82	LQTLPANPP		I84V, I84T, A86V, A86T, V88M, V88L, A90P, A90V, P92L	I73M, L75H, L77M, L77V, A79E
34–42	FGANSNNPD		23–30	FRANTANP ^¥^	*0401	-	R24K
16–24	LSVPNPLGF		5–13	LSTSNPLGF ^¥^	*0401, *1302	-	-
24–32	FFPDHQLDP		13–21	FFPDHQLDP	*0401,	F25L	-
63–71	FGPGFTPPH		52–60	FGLGFTPPH	*0101, *0401, *0701, 5*0101	F67L	L54M, L54P, F56L
34–42	FGANSNNPD		23–30	FRANTANPD	*0101, *0401, *0802, *1302	-	R24K
67–75	FTPPHGGVL		56–64	FTPPHGGLL	*0101, *0701,	F67L, G73R	-
28–36	HQLDPAFGA		17–25	HQLDPAFRA	*0301, *0401,	-	R24K
84–92	ILATVPAVP	*0101, *0301, *0401, *0701, *0802, *1101, *1302, *1501, 5*0101	73–81	ILQTLPANP	*0101, *0301, *0401, *0701, *0802,*1302,	I84V, I84T, A86V, A86T, V88M, V88L, A90P, A90V, P92L	I73M, L75H, L77M, L77V, A79E
85–93	LATVPAVPP	*0401, *0802, 5*0101	74–82	LQTLPANPP	*0802, 5*0101	I84V, I84T, A86V, A86T, V88M, V88L, A90P, A90V, P92L	I73M, L75H, L77M, L77V, A79E
74–82	LLGWSPQAQ	*1501	63–71		*0101, *0401, *0802, *1501	W77stop, S78R, P79S, A81S, A81P	S67N, P68S
16–24	LSTSNPLGF		5–13		*0301, *0701, *1302	-	-
12–21	MGTNLSVPN	*0401, *0802, *1302	1–10	MGQNLSTSNP		-	G2E
15–22	NLSVPNPLG	*0401,	4–12	NLSTSNPLGF		-	-
83–90	QGILATVPA	*0101, *0802	71–79	QGILQTLPA	*0101,	G83D, I84V, I84K, A86T, A86V, V88L, V88M, A90T, A90V	I73M, L75H, L77M, L77V, A79E
14–22	TNLSVPNPL	*0101, *0701, *1302	3–11	QNLSTSNPL	*0101, *0401,	-	-
4–12	WSAKPRKGM	*1101, 5*0101	-			S5P, A6S, A6T, K10N, K10I	-
77–85	WSPQAQGIL	*0101, *0701,	66–75	WSPQAQGIL	*0101, *0701,	W77stop, S78R, P79S, A81P, A81S, G83D, I84V, I84K	P68S,

*0101 means HLA class II allele DRB1*0101 e.tc. 5*0101 means HLA class II allele DRB5*0101. ^¥^ indicates SB.

**Table 7 viruses-12-00731-t007:** Summary of *PreS2* epitopes binding to respective alleles, and mutations that exist within the Botswana HBV sequences.

Genotype	AA Sequence Core	AA Sequence of *epi*	Pres2 *epi* Residues	Count of *epi* with Core AA Sequence	HLA Class II Alleles	Mutations in the Core Sequence
A	ALQDPRVRG	AFHQALQDPRVRGLYFPA	7–24	7	*0301^∆^	R16K, V17I
A	ASHISSISS	VPNTASHISSISSRT	35–49		*0401	A39T, I45S
D	ASPLSSIFS	VPTTASPLSSIFSRIG	35–50	2	*0101, *0401	L42I, L42S
A	FHQALQDPR	NSTAFHQALQDPRVRG	4–19	2	*0301, 5*0101	A11T, R16K
D	FHQTLQDPR	NSTTFHQTLQDPRVR	4–18	1	*0301	-
D	FSRIGDPAL	SPLSSIFSRIGDPALN	40–55	1	*0101, *0401, *0701, 5*0101	R48T, P52H, A53V, L54P
A	HISSISSRT	VPNTASHISSISSRTG	35–50	4	*0101, *0401, *0701	I45S, R48T
D	IFSRIGDPA	SPLSSIFSRIGDPALN	40–55	2	*0802, *1302, *1501	L42I, L42S
A	ISSISSRTG	PNTASHISSISSRTGDPALN	36–55	6	*0101, *0401, *0701, *0802, *1101, *1501, 5*0101	I45S, R48T
A	LNPVPNTAS	SSSGTLNPVPNTASHISSI	27–45	5	*0401, *0802	L32H, L32R, P34L, N37H, N37T, I38T, A38T
D	LSSIFSRIG	PTTASPLSSIFSRIGDPALN	36–55	6	*0101, *0401, *0701, *0802, *1101, *1501, 5*0101	P52L, A53V
A	MQWNSTAFH	MQWNSTAFHQALQDP	1–15	1	*1302	Q2I, A7T
		MQWNSTTFHQTLQDP		-	-	S5F, S5Y, T6A, T7I, T7N
D	PLSSIFSRI	VPTTASPLSSIFSRI	35–49	1	*0701	L42I, L42S
A	PVPNTASHI	SSGTLNPVPNTASHISSI	27–45	6	*1302	P34L, N37H, N37T
D	PVPTTASPL	VNPVPTTASPLSSIF	32–46	2	*0701	P34H, P36L, L42I, L42S
A	QALQDPRVR	TAFHQALQDPRVRGLYF	6–22	3	5*0101	-
D	QTLQDPRVR	STTFHQTLQDPRVRGLYF	5–22	4	5*0101	R22K
D	TLQDPRVRG	STTFHQTLQDPRVRGLYFPA		6	*0301	R18K, G19D, G19A
A	VRGLYFPAG	DPRVRGLYFPAGGSSSG	14–30	3	*0802, *1101, *1501	V17I, Y21N, Y21S, F22L, F22P, F22T, P23A
D				3	*0802, *1101, *1501	R18K, G19D, G19A, F2222Q, F22H, F22P, F22L
A	WNSTAFHQA	MQWNSTAFHQALQDP	1–15	1	*0401, *0701	A7T, A11T
D	WNSTTFHQT	MQWNSTTFHQTLQDP		1	*0701	S5F, S5Y, T6A, T7I, T7N
A	YFPAGGSSS	PRVRGLYFPAGGSSSGTLNP	15–34	6	*0101, *0401, *1101, 5*0101	Y21N, Y21S, F22L, F22P, F22T, P23A, S29L
D	YFPAGGSSS	PRVRGLYFPAGGSSSGTVNP		6	*0101, *0401, 5*0101, *1101	F22Q, F22H, F22P, F22L

*0101 means HLA class II allele DRB1*0101 e.tc. 5*0101 means HLA class II allele DRB5*0101.

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
