# Peer review of "In Silico Prediction of Human Leukocytes Antigen (HLA) Class II Binding Hepatitis B Virus (HBV) Peptides in Botswana"

_viruses, 2020, doi:10.3390/v12070731_

Round 1
Reviewer 1 Report
The authors have carefully revised the manuscript and provided enough descriptions for all figures and tables.
Reviewer 2 Report
Choga et. el. used in-silico methods to infer 15-mer peptides, based on HBV genome sequences extracted from GeneBank, that can bind to HLA class II alleles commonly expressed in the Botswana population, and studied them in detail.
This revised version of the manuscript is well written and have addressed all my previous concerns. I have no further comments.
Reviewer 3 Report
In recent years, bioinformatics has greatly facilitated vaccine design as it is safer, more convenient, more efficacious, and less expensive than traditional vaccines selection. The current study is of such kind, in which the author selected HBV surface antigen and HLA class II for peptide design. The author concluded in the end that they have identified antigenic and genotype-specific peptides suitable for vaccine as well as diagnostic kits development. However, to the reviewer, the study is still preliminary, and the conclusion not only poorly supported by the results but also failed to fulfill the stated aim.
First, the paper is not well written. Poor logic flow, language redundancy and multiple grammar mistakes made the reading very difficult.
The major scientific concerns include:
- It is not clear why surface antigen and HLA II were selected for the study (why not core protein or B cell epitope or HLA I)? It looked as if the author wanted to focus on improving the design of preventive vaccine. However, aa mutation in the context of failed HBV clearance which mediated mainly by CD8 were mentioned a lot, making the rationale of the study design unclear.
- The author should provide immunogenicity predication of the predicted epitopes.
- The toxicity analysis of the predicted epitope is insufficiently based on BLAST search of NCBI protein database (mainly sequence match). More advanced method should be used as the recognition of host tissue by the specific immune response induced by the epitope involves physical-chemical properties of the two.
- Regarding the identified candidate vaccine, how is their molecular docking to HLAII like? Can the author verify experimentally if they can form stable HLA-peptide complex in vitro?